# READ: RECURRENT ADAPTATION OF LARGE TRANS-FORMERS

## ABSTRACT

Fine-tuning large-scale Transformers has led to the explosion of many AI applications across Natural Language Processing and Computer Vision tasks. However, fine-tuning all pre-trained model parameters becomes impractical as the model size and number of tasks increase. Parameter-efficient transfer learning (PETL) methods aim to address these challenges. While effective in reducing the number of trainable parameters, PETL methods still require significant energy and computational resources to fine-tune. In this paper, we introduce **RE**current **AD**aption (READ) — a lightweight and memory-efficient fine-tuning method — to overcome the limitations of the current PETL approaches. Specifically, READ inserts a small RNN network alongside the backbone model so that the model does not have to back-propagate through the large backbone network. Through comprehensive empirical evaluation of the GLUE benchmark, we demonstrate READ can achieve a $56\%$ reduction in the training memory consumption and an $84\%$ reduction in the GPU energy usage while retraining high model quality compared to full-tuning. Additionally, the model size of READ does not grow with the backbone model size, making it a highly scalable solution for fine-tuning large Transformers.

## 1 INTRODUCTION

Large-scale transformers architecture have achieved state-of-the-art results in several Natural Language Processing (NLP) tasks Bao et al. (2021); Brown et al. (2020); Liu et al. (2019); Lu et al. (2019); Raffel et al. (2020); Wei et al. (2021). Scaling up the size of these models has been shown to confer various benefits, such as improved model prediction performance and sample efficiency Chung et al. (2022); Howard and Ruder (2018); Wei et al. (2023). The conventional paradigm is to pre-train large-scale models on generic web-scale data and fine-tune the models to downstream tasks. However, fine-tuning these models has become prohibitively expensive.

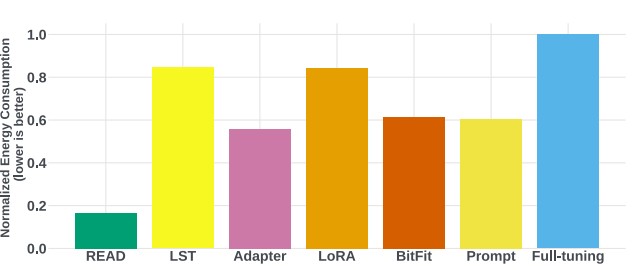

Figure 1: The normalized energy consumption relative to full-tuning on GLUE tasks.

Since 2018, the model size has increased by almost two orders of magnitude faster than GPU memory Lialin et al. (2023), resulting in prohibitively high cost to advance AI technologies Wu et al. (2022). Only a few well-funded institutions have the resources to fine-tune these models. Parameter-efficient transfer learning (PETL) Aghajanyan et al. (2020); Houlsby et al. (2019); Hu et al. (2021); Karimi Mahabadi et al. (2021); Lester et al. (2021b); Li and Liang (2021); Zaken et al. (2022) has emerged as a promising solution to overcome the challenges of full fine-tuning. Parameter-efficient transfer learning techniques aim to address these challenges by leveraging smaller and more task-specific models to efficiently adapt the pre-trained model's parameters to the target task.

However, all these methods either come with additional inference latency Houlsby et al. (2019) or reduces only a small amount of memory requirement during training — the primary motivation of PETL. Figure 1 illustrates that parameter-efficient methods, while tuning only a small percentage of the overall parameters, still consume significant energy to fine-fine. Since the updated parameters are inside the backbone language models, to calculate gradients for these parameters for backpropagation, PETL methods need to run the backward pass through the sizeable pre-trained language models. This prevents PETL methods from being applied to many real-world applications with limited computational resources.

Recent works of Side-Tuning Zhang et al. (2020a) and Ladder-Side Tuning (LST) Sung et al. (2022) propose to use a side network that takes intermediate activations from the backbone networks to reduce the need to backpropagate through the large backbone layer. It thus reduces training memory requirement. However, both Side-Tuning and LST have significant drawbacks. In Side-Tuning, the side network only consumes the original inputs, leaving the informative intermediate results from the backbone unused. LST overcomes this problem by using a side Transformer. However, Transformers are challenging to train Liu et al. (2020). Moreover, LST requires an extra pretraining stage to extract a sub-Transformer from the backbone and use it to initialize the side network, increasing the cost of fine-tuning. Additionally, the size of the side-Transformer and the backbone increase, making this approach hard to scale (Figure 5).

To overcome these challenges, we introduce REcurrent ADaption (READ), a lightweight parameter and memory efficient fine-tune method that inserts a small RNN network to the side backbone model. We show that READ can achieve comparable model quality to fine-tuning while saving more than 84% on energy consumption.

**Our contributions** The key contributions of this work are summarized as follows:

- We overcome the limitations of PETL and side-tuning methods by proposing **RE**current **AD**aptation (READ), a simple yet effective side-tuning design that requires no pretraining of the side network — a prerequisite of prior side-tuning techniques.

- We conduct thorough experiments on various NLP benchmarks, showcasing the strong performance and high efficiency of READ. On the GLUE benchmark, READ achieves competitive accuracy compared to a range of fine-tuning approaches while reducing the model training memory consumption by $56\%$ and GPU energy usage by $84\%$ relative to full-tuning (Figure 2), at almost no costs of inference latency and memory (Figure 6).

- We demonstrate that READ is a highly scalable solution to fine-tune large transformers and is independent of the backbone model size (Figure 5).

- We provide theoretical justification on why READ utilizes the backbone hidden state to perform side-tuning (Section 2.1).

## 2 BREAKING DOWN REcurrent ADaptation (READ)

### 2.1 WHAT IS READ?

Figure 3 demonstrates the mechanism of READ fine-tuning on an encoder-decoder transformer backbone $\mathcal{T}$. We freeze $\mathcal{T}$ throughout training, and initialize a trainable neural network named READ at both encoder and decoder. The major component of READ is a standard RNN, together with a *Joiner* network where multiple sources of information join to produce the inputs for RNN. During a forward pass, we first run through $\mathcal{T}$ independently from READ, and cache necessary intermediate results at every transformer layer. Next, we iteratively compute the RNN hidden states at encoder and then decoder. Lastly, we add the outputs of RNN and $\mathcal{T}$ to obtain the new final state.

We summarize the following key properties of the proposed READ network as follows:

- Forward pass is completely separated from the backbone $\mathcal{T}$. This way, backward propagation will never flow through $\mathcal{T}$, hence reducing the training memory needed for caching non-linear activations of $\mathcal{T}$ Sung et al. (2022).

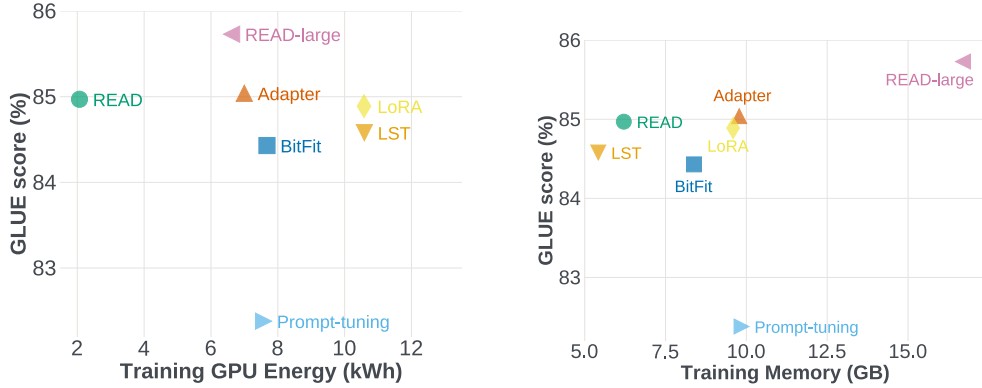

Figure 2: Comparing READ and other fine-tuning methods over GLUE tasks on training energy and peak training memory relative to full-tuning. The y-axis is the average metric over 7 GLUE tasks. (left) The x-axis is the cumulative training energy in KWh. (right) The x-axis is the GPU peak training memory during training.

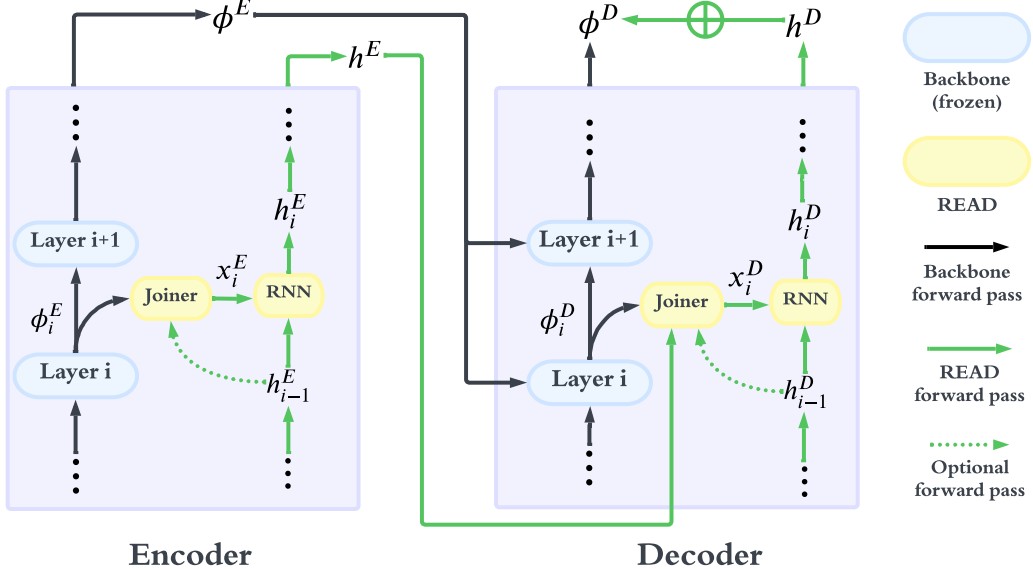

Figure 3: READ Fine-Tuning Mechanism.

- Only RNNs and feed-forward networks (FFNs) are involved, with no attention mechanism. This improves usability and training efficiency because it requires no pre-training or pruning. READ is ready to be plugged and used.

- Because of the recurrent nature of READ, the number of trainable parameters does not increase with backbone layers. The number of trainable model parameters grows sub-linearly along with the backbone size.

- Consumes without modifying the intermediate results from backbone [1].

---

[1] One advantage, which is beyond the scope of this paper, is that we can *multi-task* with multiple READ networks but only needs one pass through the backbone, reducing training costs and inference time.

## 2.2 How does READ work?

Let us begin by understanding what READ actually learns. To simplify our terminologies, let $\mathcal{L}_1, \cdots, \mathcal{L}_N$ be transformer layers in the backbone $\mathcal{T}$, and $\phi_i = (\phi_i^1, \cdots, \phi_i^m)$ be the output hidden states at $\mathcal{L}_i$ for given inputs $X$ of length $m$. Many fine-tuning methods directly modify $\phi_i$, either through updating the backbone weights, such as full tuning and partial tuning, or via injecting learnable parameters into the middle of the backbone, like Adapter, LoRA, Prompt tuning, etc. On the other hand, **READ learns the *correction* to $\phi_i$ needed for a new task**.

**Definition 2.1** (*Correction*). Let $\mathcal{T}'$ be a perturbation of $\mathcal{T}$, and $\phi_i'$ be the hidden states at layer $\mathcal{L}_i'$ of $\mathcal{T}'$. We define $\phi_i' - \phi_i$ to be a *correction* to $\phi_i$, and denote it by $\delta\phi_i$.

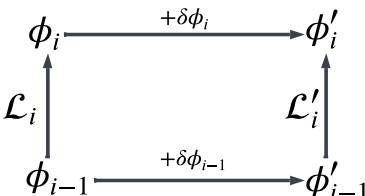

Figure 4: Commuting diagram for *correction*.

By Definition 2.1, the diagram in Figure 4 commutes. Indeed, we will show under appropriate assumptions that if $\mathcal{T}'$ is a fine-tuned (e.g. with Adapter or LoRA) version of $\mathcal{T}$, then the following equation systems gives the first order approximation $h_i$ of the true *correction* $\delta\phi_i$:

$$\begin{cases} \psi_i^\alpha = \Phi_i \cdot \mathcal{F}_i h_{i-1}^\alpha + \sum_{\beta=1}^m \sigma_i^{\alpha\beta} \Psi_i \cdot \mathcal{F}_i h_{i-1}^\beta \\ x_i^\alpha = [\phi_i^{\alpha T}, \psi_i^{\alpha T}]^T \\ h_i^\alpha = \mathcal{G}_i(\mathcal{H}_i x_i^\alpha + h_{i-1}^\alpha) \end{cases} \tag{1}$$

for $\alpha = 1, \cdots, m$.[2] Here, $\sigma_i^{\alpha\beta}$ and $\phi_i^\alpha$ are cached attention scores and hidden states, $\mathcal{F}_i, \mathcal{G}_i, \mathcal{H}_i$ are (non-linear) transformations on $\mathbb{R}^d$, and $\Phi_i, \Psi_i$ are matrix-valued linear functions taking only cached values from $\mathcal{L}_i$. Most importantly, (1) does not involve attention mechanism, as all operations only act on the column space of $\phi$. The major step of deriving (1) is to extract an inductive formula for the *corrections* $\delta\phi$ from the following identity, an equivalent form of Figure 4:

$$\mathcal{L}_i(\phi_{i-1}) + \delta\phi_i = \mathcal{L}_i'(\phi_{i-1} + \delta\phi_{i-1}). \tag{2}$$

We leave the math details to Appendix A.

In practice, we rely on a neural network — READ — to model the equation system (1) (i.e. Figure 3), which

- uses a simple *Joiner* network (combining the first two equations of 1) to compute $x_i$; we substitute $\mathcal{F}_i, \mathcal{G}_i, \mathcal{H}_i, \Phi_i, \Psi_i$ in (1) with either FFNs or linear layers and fuse the learnable parameters across all indices $i$s for parameter efficiency and scalability;

- adopts RNN to model the last equation of (1).

Layer normalization is omitted as READ uses already normalized hidden states. The down-projection matrix connects the transformer's outputs to the RNN's inputs and is within the RNN. The RNN's outputs are up-projected and added to the backbone's last hidden state (see Algorithm 1). The final loss function is cross-entropy between logits and gold labels. For further modeling details in READ experiments and a comprehensive forward algorithm of READ, refer to Appendix B and C

---

[2]The equation system (1) only applies to self-attention, and we will derive a similar formula for decoder where cross-attention is present in Appendix A.

## 3 EXPERIMENT SETUP

### 3.1 BASELINE AND OTHER STATE-OF-THE-ART DESIGNS

We compare READ against full tuning and other commonly-used PETL methods.

**Full tuning** is not an efficient fine-tuning method but serves as a strong baseline for performance.

**BitFit Ben-Zaken et al. (2021)** tunes only bias terms of the model during training.

**Prompt-tuning Lester et al. (2021a)** inserts trainable prompt vectors to the inputs' embedding vectors.

**Adapters Houlsby et al. (2019)** appends a small residual MLP after every attention and feed-forward layer. We experiment with the sequential adapter version by Houlsby et al. (2019).

**LoRA Hu et al. (2021)** inserts trainable low-rank matrices into each layer of the backbone Transformer model to parameterize the weights' changes.

**LST Sung et al. (2022)** hierarchically adds multiple side networks, with each side network responsible for modulating the activations of a specific layer in the pre-trained model.

For all PETL methods and READ, we keep the backbone transformer frozen throughout the training and only update the new parameters.

### 3.2 DATASETS

We evaluate READ and the baselines on the GLUE Wang et al. (2018) benchmarks. These benchmarks cover a variety of NLP tasks, including linguistic acceptability (CoLA Warstadt et al. (2018)), paraphrase detection (MRPC Dolan and Brockett (2005), QQP Chen et al. (2017), STS-B Cer et al. (2017)), natural language inference (MNLI Williams et al. (2017), QNLI Rajpurkar et al. (2016)), and sentiment classification (SST-2)[3]. In GLUE, the original test set labels are not publicly available. Instead, we follow Zhang et al. (2020b) and Karimi Mahabadi et al. (2021) to create a test set for each task as follows: if the training set contains less than 10k samples, we equally split the original validation set into two subsets and treat them as new validation and test sets; otherwise, we use the original validation set as the test set, and split 1k from the training set as the new validation set. For MNLI, we use the mismatched set as the validation set and the matched set as the test set. We report the dataset sizes in Appendix C.2.

### 3.3 MODEL SPECIFICATION AND EXPERIMENTAL DETAILS

We adopt the encoder-decoder T5 Raffel et al. (2019) model as our backbone transformer. We use T5$_{\text{BASE}}$ for all of our experiments, and also use T5$_{\text{LARGE}}$ for READ experiments, which we denote by READ-large. We perform fine-tuning on each dataset for up to 30 epochs and do an early stop once validation metric stops improving. For READ, we experiment with $\{128, 256\}$ as RNN hidden dimensions, $\{$RNN, LSTM, GRU$\}$ as RNN architectures. For PETL baselines, we experiment with $\{32, 64\}$ as Adapters' bottleneck sizes, $\{8, 32\}$ as LoRA's ranks, and $\{10, 20, 30\}$ as Prompt-tuning's prompt sizes. For all experiments, we conduct a grid search for learning rates in between $[1 \times 10^{-6}, 3 \times 10^{-3}]$ on log scale for up to 32 rounds. We choose the hyperparameters that achieve the best validation scores and report their test scores. Complete setup and hyperparameters detail are in Appendix C.3.

### 3.4 ENERGY CONSUMPTION MEASUREMENT

Higher training efficiency translates to lower energy consumption. To demonstrate the training efficiency benefit of READ, we measure and report the training GPU energy consumption (in kWh) for every experiment. We adopt the following commonly-used methodology to measure and estimate the model training energy consumption. We take the GPU resource utilization into account when computing the corresponding energy consumption by assuming a simple linear relationship between GPU utilization and its power consumption. Assume a training experiment endures $H$ hours on

---

[3]We exclude RTE from GLUE due to its small size compared to other tasks

GPUs, with power consumption of $p_0$ kW, at the GPU utilization level (summed over all GPU nodes) $u(t)$ (in percent). Then the total energy consumption (in kWh) is given by

$$E = \int_0^H \frac{u(t)}{100} \cdot p_0 dt = H \cdot \left( \frac{1}{H} \int_0^H u(t) dt \right) \cdot \frac{p_0}{100}. \tag{3}$$

In practice, we sample $u(t)$ at the granularity of minutes throughout training using NVIDIA's System Management Interface (smi). We then calculate its cumulative sum $\overline{U} = \sum_{i=1}^{60H} u_i$, thereby we can approximate the right hand side of Equation (3) by

$$H \cdot \frac{\sum_{i=1}^{60H} u_i}{60H} \cdot \frac{p_0}{100} = \overline{U} \cdot \frac{p_0}{6000}. \tag{4}$$

When reporting the energy consumption analysis for READ and other designs (see Section 4), we use $p_0 = 0.25$ kW for a NVIDIA V100 32 GB GPU [4] for Equation (4).

## 4 EVALUATION RESULTS

We train and evaluate each method on all the GLUE tasks. We take the cumulative energy consumption and measure the peak GPU during training. In this section, we report and analyze the results on the GLUE Benchmarks. Every method other than READ in this Section is not memory-efficient, and we postpone the comparison with LST to Appendix **??** due to its memory-efficient nature.

**READ outperforms other methods while consuming significantly lower energy:** Figure 2 (left) shows that READ can reduce GPU energy consumption by up to 90% compared to full-tuning. READ lowers the GPU memory footprint by 56% while retaining the same model accuracy when retraining. While other parameter-efficient transfer learning (PETL) methods like LoRA, BitFit or Adapter reduce the number of trainable parameters, they do not reduce the compute cost required to fine-tune. We believe the underlying optimization objective for PETL is to reduce this compute cost. Table 1 shows the performance of all methods on GLUE with T5BASE. Excluding Adapter, READ outperforms all parameter-efficient methods while consuming at least 68% less energy. Compared to Adapter, READ achieves nearly the same model accuracy (less than 0.1% lower) while using 70% less energy. More interestingly, READ with T5LARGE (i.e. READ-large) achieves better performance than all other methods and consumes similar or less energy compared to other methods. For example, READ-large outperforms Full-tuning and Adapter by 1.4% and 0.8% with 69% and 5% less energy, respectively. These results show that by using READ, we can scale up the model size while keeping the same hardware and memory constraints.

**READ consumes less training memory:** Figure 2 (right) shows the design space trade-off between model quality performance and memory footprint. READ improves the training memory requirement by at least 25% compared to all the other baselines while achieving similar or better performance. READ with $T5_{\text{LARGE}}$ consumes similar amount of memory as full-tuning with $T5_{\text{BASE}}$. As the backbone size increases, the memory savings achieved by READ become increasingly significant in comparison to the other PETL methods, as depicted in Figure 5 (right). Notably, at the $T5_{\text{3B}}$ backbone level, these savings reach as high as 43%. This observation suggests that READ is remarkably effective in the regime of fine tuning large Transformers.

**READ is scalable:** As shown in Figure 5 (left), the number of trainable parameters of READ scale more slowly as compared to the other PETL methods. READ's number of parameters exhibits a log-linear growth pattern as the T5 backbone model size increases. In fact, the recurrent nature of READ makes its tunable size independent from the number of backbone layers, making READ a more suitable choice for fine-tuning large Transformers in practice.

**READ achieves competitive inference latency and memory efficiency:** As Figure 6 (left) and Table 3 indicate, READ achieves comparable inference latency and memory requirement as the other PETL methods. To assess the inference memory impact of READ more comprehensively, we use Figure 6 (right) to demonstrate that, as the backbone size increases, the inference memory growth (relative to Full-tuning) of READ becomes less noticeable and decays to a similar extent as the other methods at $T5_{\text{LARGE}}$.

---

[4] 250W comes from the datasheet on NVIDIA's website

Table 1: Performance and energy consumption results of all methods on GLUE tasks. We report the accuracy for SST-2, MNLI, QNLI, and Matthew's Correlation for CoLA. For STS-B we report the average of Pearson correlation and Spearman correlation. For MRPC and QQP, we report the average of F1 score and accuracy. For all tasks, we report the average score on 3 different seeds. Bold fonts indicate the best results of that column.

| Method | Trainable Params (%) | Power (kW) | Energy (kWh) | CoLA | MNLI | QNLI | MRPC | QQP | SST-2 | STS-B | Avg. |
|---|---|---|---|---|---|---|---|---|---|---|---|
| *Baselines* | | | | | | | | | | | |
| Full-tuning | 100 | 0.77 | 12.52 | 53.97 | 86.17 | 90.87 | 86.88 | 89.71 | 92.89 | 88.19 | 84.52 |
| Adapter | 0.96 | 0.50 | 6.99 | 52.56 | 85.68 | 92.89 | 87.84 | 88.95 | 93.12 | 87.51 | 85.04 |
| LoRA | 0.48 | 0.68 | 10.58 | 51.70 | 85.20 | 92.72 | **88.07** | 88.92 | 93.46 | 86.97 | 84.89 |
| BitFit | 0.06 | 0.47 | 7.68 | 50.92 | 85.28 | 92.58 | 86.32 | 88.70 | **94.15** | 86.94 | 84.43 |
| Prompt-tuning | **0.01** | 0.50 | 6.45 | 42.71 | 79.38 | 91.73 | 86.04 | 88.74 | 93.12 | 84.96 | 82.38 |
| LST | 2.00 | 0.44 | 10.59 | 53.38 | 84.53 | 92.43 | 87.38 | 88.31 | 92.09 | 87.37 | 84.58 |
| *Our method* | | | | | | | | | | | |
| READ | 0.80 | **0.43** | **2.06** | 52.59 | 85.25 | 92.93 | 87.09 | 89.10 | 93.80 | 87.77 | 84.97 |
| READ-large | 0.32 | 0.62 | 6.62 | **54.05** | **87.29** | **93.68** | 87.70 | 89.34 | **93.92** | **88.58** | **85.73** |

Table 2: READ with and without recurrency

| Method | CoLA | MNLI | QNLI | MRPC | QQP | SST-2 | STS-B | Avg. | Trainable Params (%) |
|---|---|---|---|---|---|---|---|---|---|
| READ (with Recurrency) | 52.59 | 85.25 | 92.93 | 87.09 | 89.10 | 93.80 | 87.77 | 84.97 | 0.8 |
| READ (w/o. Recurrency) | 53.24 | 84.10 | 91.51 | 89.02 | 89.18 | 94.04 | 87.10 | 85.15 | 9.6 |
| READ-large (with Recurrency) | 54.05 | 87.29 | 93.68 | 87.70 | 89.34 | 93.92 | 88.58 | 85.73 | 0.32 |
| READ-large (w/o Recurrency) | 50.17 | 86.90 | 93.00 | 87.61 | 89.12 | 94.15 | 87.46 | 85.02 | 6.4 |

**The importance of recurrency** We perform ablation analysis on the importance of recurrence in READ in Table 2. We find that the removal of recurrence does not significantly enhance READ quality and even diminishes quality for the T5 large backbone. However, without recurrence leads to over 12 times more trainable parameters, compromising scalability.

**Comparison with Ladder-Side-Tuning (LST)** We compare our methods with Ladder-Side-Tuning (LST), another memory efficient fine-tuning approach Sung et al. (2022). We follow the pruning method introduced in Sung et al. (2022) to extract a smaller transformer from the backbone transformer and use it to initialize the side transformer, and re-implement LST. Table **??** lists the results of LST (using T5$_{BASE}$) on GLUE benchmarks and its energy efficiency. The results indicate that READ (base) outperforms LST (base) on most tasks (except for a tiny task MRPC), using **80% less energy consumption** and **60% less trainable parameters**. While LST consumes $15\%$ less peak training memory relative to READ, it takes **40% more inference memory** and **77% longer inference time** than READ, a consequence of its attention-based side-network architecture. It is also noteworthy that when compared to LST even READ-large saves $38\%$ GPU energy and yields a similar inference latency, with $1.4\%$ relative gain on the averaged GLUE score. Furthermore, the "pre-training stage" refers to the process described in LST paper section 2.2, where distillation is performed with T5 pre-training objective. It is important to note that caching the attention outputs does not involve updating any model parameters and should not be considered as a form of training.

Table 3: Average inference memory consumption (GB) for every method with different backbones on GLUE benchmark.

| | READ | LST | Adapter | LoRA | Prompt | Bias | Full |
|---|---|---|---|---|---|---|---|
| T5$_{SMALL}$ | 0.317 | 0.427 | 0.303 | 0.302 | 0.301 | 0.301 | 0.300 |
| T5$_{BASE}$ | 0.966 | 1.358 | 0.952 | 0.948 | 0.948 | 0.945 | 0.943 |
| T5$_{LARGE}$ | 2.943 | 4.597 | 2.936 | 2.925 | 2.925 | 2.914 | 2.912 |
| T5$_{3B}$ | 10.885 | 11.400 | 10.878 | 10.866 | 10.894 | 10.855 | 10.853 |

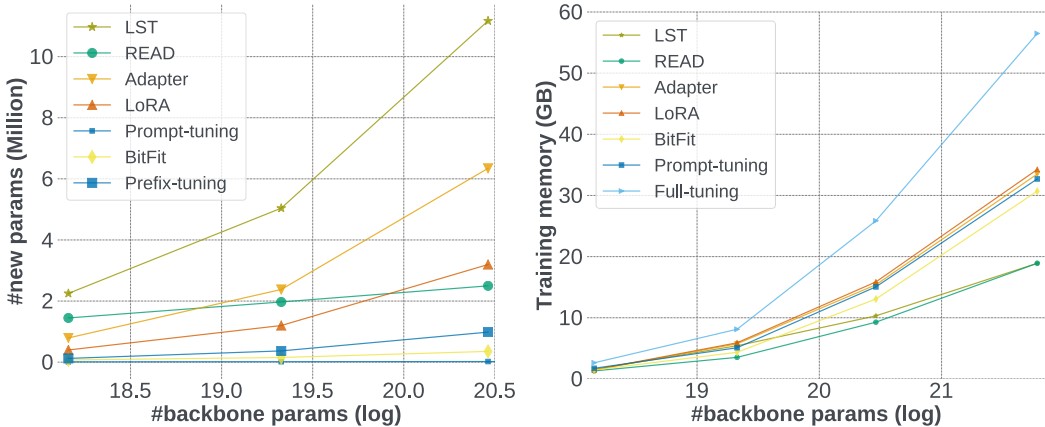

Figure 5: (left) The number of trainable parameters as the backbone model size increases. (right) The peak training memory as the backbone model size increases. For backbone models we use T5$_{\text{SMALL}}$, T5$_{\text{BASE}}$, T5$_{\text{LARGE}}$. For the memory plot (right) we also include T5$_{\text{3B}}$, and use batch size $48$ on MNLI.

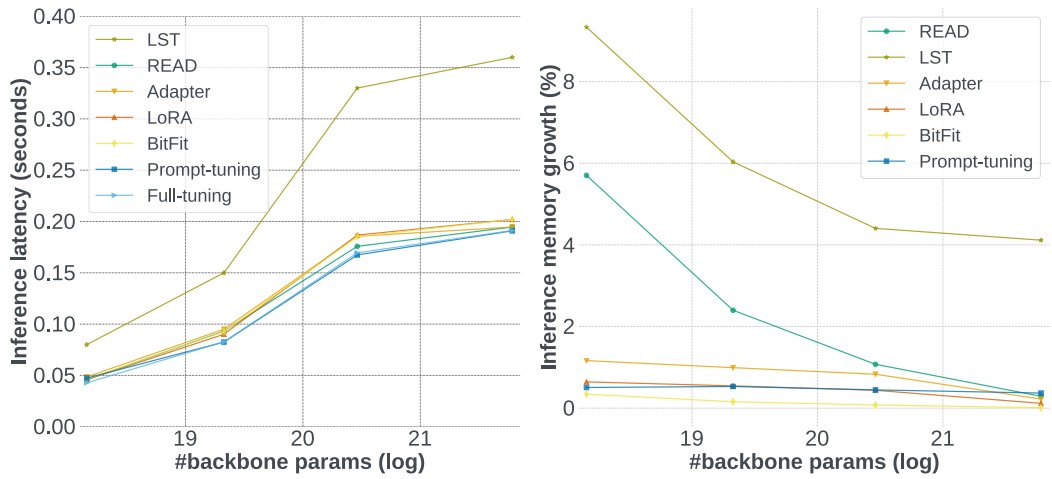

Figure 6: (left) Inference latency as backbone model size increases. (right) Inference memory growth (relative to full-tuning) in percentage as backbone model size increases (all methods have very similar inference memory and we have to use a percentage plot to distinguish them). In both figures, we use T5$_{\text{SMALL}}$, T5$_{\text{BASE}}$, T5$_{\text{LARGE}}$, and T5$_{\text{3B}}$ as backbones.

## 5 RELATED WORK

**Parameter-efficient Transfer Learning.** There has been an explosion of generative AI applications in recent months Biderman et al. (2023); Rombach et al. (2022); Touvron et al. (2023); Wei et al. (2021). However, the ability to fine-tune large transformers is primarily limited by the growing compute cost required to adapt and serve these models. Parameter-efficient transfer learning (PETL) Aghajanyan et al. (2020); Houlsby et al. (2019); Lester et al. (2021b); Li and Liang (2021); Lialin et al. (2023); Sung et al. (2022); Zaken et al. (2022) aims to solve this problem by training only a small set of parameters. There are many PETL methods which we defer the reader to Lialin et al. (2023) for a more comprehensive overview. In this section, we will summarize the most popular PETL methods which we used as baselines. Adapter-based approaches He et al. (2021); Houlsby et al. (2019) insert small learnable modules between pre-trained model layers and only update these adapters during fine-tuning, reducing computational cost and memory requirements. Low-Rank Adaptation (LoRA)

Hu et al. (2021) injects trainable rank decomposition matrices into each layer of the Transformer model. BitFit Zaken et al. (2022) fine-tunes only the biases of the model. Prompt-tuning Lester et al. (2021b) is a successor of Prefix-Tuning Li and Liang (2021), which adds a continuous task-specific prompt to the input. In contrast, current PETL approaches aim to minimize the number of parameters trained. These approaches do not lead to memory efficiency, a more meaningful objective than parameter efficiency. This work proposes READ, simple memory-efficient methods by inserting a small recurrent network into the backbone.

**Memory-Efficient Training.** Memory-efficient training reduces memory consumption by reducing the storage of intermediate activations Sung et al. (2022). Gradient checkpointing Chen et al. (2016) reduces memory consumption during backpropagation by storing a subset of intermediate activations and recomputing them as needed, trading time for memory. Reversible layers Gomez et al. (2017) reconstruct each layer's activations from the next layer's activations. ZeRO Rajbhandari et al. (2020) partitions model states, gradients, and optimizer states across multiple devices for distributed training, significantly reducing memory redundancy. Layer-wise Adaptive Rate Scaling (LARS) You et al. (2017) dynamically scales learning rates for different layers, reducing memory overhead associated with large gradients and enabling the training of large models with limited memory.

**Sidenet Tuning.** Side-tuning Zhang et al. (2020a) adds a lightweight side network alongside the pre-trained model. During training, the side network and the task-specific head are updated while the pre-trained model's parameters are kept fixed. The side network learns to modulate the pre-trained model's activations, allowing it to adapt to the new task without altering the base model. Ladder side-tuning Sung et al. (2022) hierarchically adds multiple side networks, with each side network responsible for modulating the activations of a specific layer in the pre-trained model. While READ takes inspiration from Side-Tuning and LST, we would like to highlight significant differences between READ and prior works. First, READ only contains a single RNN block which takes in the hidden state of the backbone network in a recurrent manner. This way, the number of parameters to fine-tune does not increase with the size of the backbone, whereas LST attaches multiple transformer blocks to the backbone network. When the backbone gets larger, the size of the LST network also gets larger. Secondly, Side-Tuning uses an additive side network to sum its representation with the backbone network in only the last layer. READ consumes the backbone's hidden state at every layer to iteratively calculate its RNN states. The recurrence nature of RNN allows information to flow from one layer to the next, which is why READ outperforms other PETL methods. Last, our fine-tuning is transformer-free as only RNN and Feed-Forward Network (FNN) structures are used in READ and require no transformer or attention mechanism. We may use a randomly initialized READ network without going through pre-training like in LST or exploiting any subtle tricks for training a transformer.

## 6 CONCLUSION AND LIMITATIONS

**Limitations.** Due to our limited computing resources, we could not scale the backbone to an even larger scale. A future direction is to fine-tune READ on Llama-7B Touvron et al. (2023) or even larger variants. Another direction can be studied if READ can generalize well in a low-data regime. A drawback of READ is its tendency to require more epochs to converge on small datasets than other PETL methods. Consequently, although READ is more efficient in per-unit time computations, it may not yield significant overall consumption gains when a task has few data points. We leave investigating READ on the low-data regime as future work.

**Conclusion.** In this paper, we propose REcurrent ADaption (READ), a lightweight and efficient parameter and memory-efficient fine-tuning method, for large-scale transformers. We show that READ achieves comparable accuracy to full fine-tuning while saving more than 84% of energy consumption and reducing training memory consumption by 56% relative to full-tuning. We demonstrate the scalability of READ because READ is independent of backbone model size. We hope that READ can make fine-tuning large models more accessible to a broader range of researchers and applications.

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
