# A APPENDIX

## A.1 REVISIT TRANSFORMER

In this subsection, we briefly revisit the computation of transformer and introduce some convenient notations for the future. Let $\mathcal{T}$ be a transformer of dimension $d$ with $N$ layers $\mathcal{L}_1, \cdots, \mathcal{L}_N$. At each layer $\mathcal{L}_i$, let the feed-forward network be $\mathcal{F}_i$ and multihead-attention be $\mathcal{A}_i$. Given a context sequence of $m$ tokens, we can express each layer as a mapping from $\mathbb{R}^{m \times d}$ to $\mathbb{R}^{m \times d}$ as follows:

$$\mathcal{L}_i = (\mathcal{F}_i^* + I) \circ \mathcal{A}_i^* + I, \tag{5}$$

where $I$ represents the identity mapping, $\circ$ denotes mapping composition, and $\mathcal{F}_i^* =: \mathcal{F}_i \circ \mathrm{LN}$, $\mathcal{A}_i^* =: \mathcal{A}_i \circ \mathrm{LN}$ (i.e. compositions with layer-normalization). Further, we define $\mathcal{R}_i = (\mathcal{F}_i^* + I) \circ \mathcal{A}_i^*$ so as to write layer mapping as $\mathcal{L}_i = \mathcal{R}_i + I$.

## A.2 DERIVATIONS OF READ

Following the notations in Subsection A.1, we derive an inductive formula for the *corrections*:

$$
\begin{aligned}
\delta\phi_i &= \phi_i' - \phi_i \\
&= (\mathcal{R}_i' + I)(\phi_{i-1}') - (\mathcal{R}_i + I)(\phi_{i-1}) \\
&= \mathcal{R}_i'(\phi_{i-1}') - \mathcal{R}_i(\phi_{i-1}) + (\phi_{i-1}' - \phi_{i-1}) \\
&= \mathcal{R}_i'(\phi_{i-1}') - \mathcal{R}_i(\phi_{i-1}) + \delta\phi_{i-1} \\
&= (\mathcal{R}_{i-1}' - \mathcal{R}_i)(\phi_{i-1}') + (\mathcal{R}_i(\phi_{i-1}') - \mathcal{R}_i(\phi_{i-1})) + \delta\phi_{i-1} \\
&= \delta\mathcal{R}_i(\phi_{i-1}') + J\mathcal{R}_i\delta\phi_{i-1} + \delta\phi_{i-1}.
\end{aligned}
\tag{6}
$$

Here $\delta\mathcal{R}_i$ denotes the operator difference $\mathcal{R}_i' - \mathcal{R}_i$, and $J\mathcal{R}_i$ is the Jacobian matrix of $\mathcal{R}_i(\cdot)$ evaluated at some point lying on the line segment from $\phi_{i-1}$ to $\phi_{i-1}'$. To simplify our arguments, we (1) assume that $J\mathcal{R}_i$ takes value at $\phi_{i-1}$, (2) let $\mathcal{T}'$ be the consequence of fine-tuning with Adapter or LoRA (applied at FFN layers $\mathcal{F}_i$)[5]. We use $\mathcal{P}$ to denote a common module adopted by Adapter and LoRA which consists of a down projection matrix to a lower dimension possibly followed by a non-linear activation, and then composed with an upper projection back to the original dimension[6]. Under these assumptions, the first term of the RHS in (6) now becomes

$$
\delta\mathcal{R}_i(\phi_{i-1}') = \begin{cases} \mathcal{P}_i \circ (\mathcal{P}_i + I)^{-1}\phi_i' & \text{(Adapter)} \\ \mathcal{P}_i \circ (\mathcal{P}_i + \mathcal{F}_i)^{-1}\phi_i' & \text{(LoRA)} \end{cases}
\tag{7}
$$

$$
=: \mathcal{W}_i(\phi_i + \delta\phi_i)
$$

Now plugging (7) back to (6), we obtain

$$\delta\phi_i = \mathcal{W}_i(\phi_i + \delta\phi_i) + J\mathcal{R}_i\delta\phi_{i-1} + \delta\phi_{i-1}. \tag{8}$$

Notice that both sides of equation (8) contains $\delta\phi_i$. Because of the non-linearity of $W_i$, there is no straightforward way to extract an inductive formula of $\delta\phi_i$ from (8).

However, let us rewrite equation (8) as

$$
\begin{aligned}
\delta\phi_i - \mathcal{W}_i(\phi_i + \delta\phi_i) - (J\mathcal{R}_i\delta\phi_{i-1} + \delta\phi_{i-1}) \\
= F(\delta\phi_i, \phi_i, J\mathcal{R}_i\delta\phi_{i-1} + \delta\phi_{i-1}) = 0,
\end{aligned}
\tag{9}
$$

and compute the Jacobian to see that $J_{\delta\phi_i}F = I - J\mathcal{W}_i$, which is invertible when $\mathcal{P}_i$ (and hence $\mathcal{W}_i$ 7) has small norm. Now by Implicit Function Theorem there exists $G$ such that

$$\delta\phi_i = G(\phi_i, J\mathcal{R}_i\delta\phi_{i-1} + \delta\phi_{i-1}). \tag{10}$$

An alternative argument is to use a first order approximation of $\mathcal{W}_i(\phi_i + \delta\phi_i)$ assuming that $\delta\phi_i$ is sufficiently small, which gives us the following inductive formula:

$$\delta\phi_i = (I - J\mathcal{W}_i)^{-1} \circ \left( \mathcal{W}_i\phi_i + J\mathcal{R}_i\delta\phi_{i-1} + \delta\phi_{i-1} \right) \tag{11}$$

---

[5]For fine-tuning methods that modify attention, we expect a similar conclusion that demands a more intricate line of reasoning, which we defer to future research.

[6]The operator norm of $\mathcal{P}$ is small when its two matrices have small weights, and therefore addition with $\mathcal{P}$ will not change the invertibility of an already invertible operator.

We take the second approach above and adopt formula (11) as we move forward, because of its explicit function form. Note that every operation in (11) acts on the column space of $\phi$ except for the Jacobian transform $JR_i$, so let us first focus on expanding $JR_i\delta\phi_{i-1}$. In fact, we will compute the Jacobian for a general attention mapping that takes 3 arguments $\phi_q, \phi_k, \phi_v$ (i.e. hidden states of queries, keys, and values), and then apply the results to the special case of self-attention (as in encoder) and cross-attention (as in decoder) respectively. For the sake of brevity, we assume that the number of attention head is 1 and omit the output projection, as neither of which is essential to our conclusion.

Let $\phi_q, \phi_k, \phi_v$ be matrices in $\mathbb{R}^{m_q \times d}, \mathbb{R}^{m_k \times d}, \mathbb{R}^{m_k \times d}$, which stand for $\mathbb{R}^d$-vector representations of the query, key, and value token sequences with length $m_q, m_k, m_k$ respectively. We use an upper index $\alpha$ to denote the vector associated to the $\alpha^{\text{th}}$ token, and omit the lower layer index $i$ when no ambiguity is present (e.g. $\mathcal{A}^\alpha$ is the $\alpha^{\text{th}}$ column of $\mathcal{A}$'s output.). First, we have

$$JR_i\delta\phi_{i-1} = (J\mathcal{F}^* + I) \circ J\mathcal{A} \circ J\text{LN}(\delta\phi_{i-1}) \tag{12}$$

by chain rule. Next we expand $J\mathcal{A}$, as every other operation in (12) acts on the column space of $\phi$; especially, up to composing with a feed-forward neural network, let us replace $J\text{LN}$ by an identity to simplify our notations. A straightforward computation gives the following:

$$J_{\phi_q}\mathcal{A}^\alpha(\delta\phi_q) = \left[\sum_{\beta=1}^{m_q} \sigma^{\alpha\beta} \cdot \frac{(v^\beta - \mathcal{A}^\alpha) \cdot k^{\beta T}}{\sqrt{d}} \cdot W_Q\right]\delta\phi_q^\alpha,$$

$$J_{\phi_k}\mathcal{A}^\alpha(\delta\phi_k) = \sum_{\beta=1}^{m_k} \sigma^{\alpha\beta} \cdot \frac{(v^\beta - \mathcal{A}^\alpha) \cdot q^{\alpha T}}{\sqrt{d}} \cdot W_K\delta\phi_k^\beta, \tag{13}$$

$$J_{\phi_v}\mathcal{A}^\alpha(\delta\phi_v) = \sum_{\beta=1}^{m_k} \sigma^{\alpha\beta} \cdot W_V\delta\phi_v^\beta.$$

Here $W_Q, W_K, W_V$ denote the query, key, and value projection matrices of $\mathcal{A}$, $q^\alpha = W_Q\phi_q^\alpha$, $k^\alpha = W_K\phi_k^\alpha$, and $\sigma^{\alpha\beta} = \text{softmax}(q^{\alpha T} \cdot k^\beta/\sqrt{d})$.

**Case 1, $\mathcal{A}$ is self-attention** Upon setting $\phi_q, \phi_k, \phi_v$ to $\phi$, and $\delta\phi_q, \delta\phi_k, \delta\phi_v$ to $\delta\phi$ in (13), we obtain:

$$J\mathcal{A}^\alpha\delta\phi = \left[\sum_{\beta=1}^{m_q} \sigma^{\alpha\beta} \cdot \frac{(v^\beta - \mathcal{A}^\alpha) \cdot k^{\beta T}}{\sqrt{d}} \cdot W_Q\right]\delta\phi^\alpha$$
$$+ \sum_{\beta=1}^{m_q} \sigma^{\alpha\beta} \cdot \left[\frac{(v^\beta - \mathcal{A}^\alpha) \cdot q^{\alpha T}}{\sqrt{d}} \cdot W_K + W_V\right]\delta\phi^\beta. \tag{14}$$

Note the two quantities in the square brackets are $\mathbb{R}^{d \times d}$-matrix-valued linear functions of values that can be computed from the cached results at $\mathcal{L}_i$, which we shall denote by $\Phi, \Psi$ from now on:

$$J\mathcal{A}^\alpha\delta\phi = \Phi \cdot \delta\phi^\alpha + \sum_{\beta=1}^{m_q} \sigma^{\alpha\beta}\Psi \cdot \delta\phi^\beta. \tag{15}$$

Now, upon inserting (15) to the $\alpha^{\text{th}}$-column of (12) by setting $\phi$ as $\phi_i$, and then plugging (12) back in (11), we obtain the iterative formula for outputs $h_i$:

$$\begin{cases} \psi_i^\alpha = \Phi_i \cdot \mathcal{F}_i\delta\phi_{i-1}^\alpha + \sum_{\beta=1}^m \sigma_i^{\alpha\beta}\Psi_i \cdot \mathcal{F}_i\delta\phi_{i-1}^\beta \\ x_i^\alpha = [\phi_i^{\alpha T}, \psi_i^{\alpha T}]^T \\ \delta\phi_i^\alpha = \mathcal{G}_i(\mathcal{H}_i x_i^\alpha + \delta\phi_{i-1}^\alpha) \end{cases} \tag{16}$$

where $\Phi_i, \Psi_i$ are defined as in (15), and $\mathcal{F}_i, \mathcal{G}_i, \mathcal{H}_i$ are FNNs that simulate $J\text{LN}$, $(I - J\mathcal{W}_i)^{-1}$, and $[\mathcal{W}_i, J\mathcal{F}_i^* + I]$ respectively; see (11) and (12). Note (16) is exactly (1) upon replacing $\delta\phi$ by $h$.

**Case 2, $\mathcal{A}$ is cross-attention** Since the decoder's *correction* iterative formula follows from a similar line of reasoning as self attention, we present the final results while omitting the details:

$$\begin{cases} \psi_i^\alpha = \Phi_i \cdot \mathcal{F}_i^D\delta\phi_{i-1}^{D,\alpha} + \sum_{\beta=1}^m \sigma_i^{\alpha\beta}\Psi_i \cdot \mathcal{F}_i^E\delta\phi^{E,\beta} \\ x_i^\alpha = [\phi_i^{\alpha T}, \psi_i^{\alpha T}]^T \\ \delta\phi_i^{D,\alpha} = \mathcal{G}_i(\mathcal{H}_i x_i^\alpha + \delta\phi_{i-1}^{D,\alpha}) \end{cases} \tag{17}$$

Table 4: Efficiency results of LST, READ, and Full-tuning. We report the training GPU energy usage summed over all tasks, and the peak training memory (per batch) averaged over all tasks. For inference memory/time, we use MNLI and report the average per batch (with test batch size 1).

| | Training GPU Energy (kWh) | Training Memory (GB) | Trainable Params (Million/%) | Inference Time (s) | Inference Memory (GB) |
|---|---|---|---|---|---|
| Full-tuning | 12.52 | 17.86 | 247.58/100.00 | **0.083** | **0.943** |
| LST | 10.59 | **5.77** | 5.04/2.00 | 0.165 | 1.358 |
| READ | **2.07** | 6.90 | **1.97/0.80** | 0.093 | 0.966 |
| READ-large | 6.62 | 17.74 | 11.17/1.4 | 0.175 | 2.943 |

where an upper index $D\backslash E$ are used to distinguish between the hidden states of decoder and encoder, and $\delta\phi^E$ is the final *correction* of encoder.

# B  ARCHITECTURE

## B.1  ARCHITECTURE CHOICES

The matrix functions $\Psi, \Phi$ in equation (16) and (17) requires computing dot products for $m^2$ pairs of vectors (13) with time complexity as large as $O(m^2 d^2)$. To reduce latency cost in practice, we make substantial reductions to the first equation in both (16) and (17) for READ experiments in this paper, as listed below:

- Indices $i$s are removed and learnable parameters are fused across all layers;
- In self-attention, we set $\Psi, \Phi$ to be constantly zeros; in other words, only hidden states are cached and used for encoder *corrections*;
- In cross-attention, we set $\Phi$ to zero and $\Psi \cdot \mathcal{F}_i^E h_{i-1}^\beta =: L h_{i-1}^\beta$, where $L$ is a learnable linear projection, so besides decoder hidden states we also need to cache the cross-attention scores for computing decoder *corrections*. Furthermore, we use a simple addition operation to combine $\phi_i$ and $\psi_i$ in (17) instead of a learnable layer.

Note some reductions we made above might be over-simplified but this paper does not explore other more sophisticated[7] while still computationally efficient options, such as a gated neural network:

$$\begin{cases} \Phi \cdot \mathcal{F}_i(h_{i-1}^\alpha) = \text{Gate}(\phi_i^\alpha) \odot \text{FFN}(h_{i-1}^\alpha), \\ \Psi \cdot \mathcal{F}_i(h_{i-1}^\beta) = \text{Gate}(\phi_i^\alpha) \odot \text{FFN}(h_{i-1}^\beta), \end{cases} \tag{18}$$

where $v \odot X = \text{diag}(v) \cdot X$. We leave the pertinent explorations to future works.

## B.2  READ ALGORITHM

Algorithm 1 outlines a forward pass during READ fine-tuning. Let $\mathcal{T}$ be a transformer with $N^E$ encoder layers and $N^D$ decoder layers, and $X\backslash Y$ be source\target sequences of length $m\backslash n$:

# C  EXPERIMENTAL DETAILS

## C.1  GPU ENERGY ANALYSIS

To provide a comprehensive understanding, we include an analysis below to show the mean and standard deviation for the sums of GPU energy, epochs to convergence, and training time across all GLUE tasks. While this analysis does reveal some variations in energy/time levels, they are not significantly substantial to alter the general trend, as READ continues to stand out as the most energy-efficient approach, with faster convergence than most baselines except for full-tuning.

---

[7]A more sophisticated choice potentially introduces more dependency on cached results and likely to improve performance at a trade-off of higher number of computation flops.

---

**Algorithm 1** READ Fine Tuning Algorithm

---

Initialize RNNs $\mathcal{N}^E, \mathcal{N}^D$ and a learnable projection $\Psi$.

$\{\phi_i^{E,\alpha}\}_{i=1,\alpha=1}^{N^E,m}, \{\phi_j^{D,\alpha}\}_{j=1,\alpha=1}^{N^D,n}, \{\sigma_i^{E,\alpha\beta}\}_{i=1,\alpha=1,\beta=1}^{N^E,m,m}, \{\sigma_j^{D,\alpha\beta}\}_{j=1,\alpha=1,\beta=1}^{N^D,n,m} \leftarrow \mathcal{T}(X,Y)$

$h_{E,0} \leftarrow 0$                         ▷ We assume embeddings need no *corrections*.

**for** $i$ in $1, \cdots, N^E$ **do**              ▷ Iteratively compute encoder *corrections*.

    $h_i^{E,\alpha} = \mathcal{N}^E(\phi_i^\alpha, h_{i-1}^{E,\alpha}), \forall \alpha$

$h_{D,0} \leftarrow 0$

**for** $j$ in $1, \cdots, N_D$ **do**             ▷ Iteratively compute decoder *corrections*.

    $\psi_j^\alpha = \sum_{\beta=1}^m \sigma_{D,j}^{\alpha\beta} \Psi h_{N^E}^{E,\beta}, \forall \alpha$

    $x_j^\alpha = \phi_j^\alpha + \psi_j^\alpha, \forall \alpha$

    $h_j^{D,\alpha} = \mathcal{N}^D(x_j^\alpha, h_{j-1}^{D,\alpha}), \forall \alpha$

$\phi_{N_D}^D{}' \leftarrow \phi_{N^D}^D + h_{N^D}^D$                   ▷ Obtain *adapted* outputs.

---

|        | Full | Adapter | LoRA | Prompt | BitFit | LST | READ |
|--------|------|---------|------|--------|--------|-----|------|
| Energy | $12.52_{0.44}$ | $6.99_{0.62}$ | $10.58_{1.9}$ | $6.45_{0.24}$ | $7.68_{0.06}$ | $10.59_{0.3}$ | $2.06_{0.18}$ |
| Epoch | $7.0_0$ | $13.01_{0.58}$ | $25.84_{3.9}$ | $34.85_{2.4}$ | $23.61_{1.3}$ | $23.58_{1.5}$ | $12.31_{1.4}$ |
| Time | $472.74_{12.77}$ | $485.2_{51.12}$ | $409.4_{19.52}$ | $315.53_{5.01}$ | $292.89_{4.03}$ | $984.55_{19.07}$ | $155.11_{8.4}$ |

Table 5: GPU Energy Consumption and Training Time across 3 trials.

## C.2 DATASET AND MODEL DETAILS

**GLUE Datasets** In Table 6, we list the dataset size, number of GPU nodes, and training batch size per GPU node for every task in GLUE. Note the total batch size (summed over all nodes) are fixed as 96 across all tasks and all methods.

**T5 models** Table 7 gives architecture-related numbers for four sizes of $T_5$ model. Note for all experiments $T5_{\text{BASE}}$ we use the original archtectures, while for READ experiments with $T5_{\text{LARGE}}$, we drop the last 4 layers from both encoder and decoder.

## C.3 HYPERPARAMETERS

**Architecture search** For fine-tuning methods that have tunable architectural hyperparameters (e.g. RNN hidden dimensions in READ, ranks in LoRA, etc), we do hyperparameter search as follows: first, we fix the architecture $\mathcal{A}$ (e.g. in READ, take RNN-dim = 128 and side-net type to be LSTM), and do learning rate search for every dataset $\mathcal{D}$. Among each hyperparameter sweep $\mathcal{H}(\mathcal{D})$ there exists a run $\mathcal{R}^*(\mathcal{D})$ that has the best validation score $\mathcal{S}(\mathcal{D})$. Then we calculate the average of $\mathcal{S}(\mathcal{D})$ across all datasets $\mathcal{D}$ as the quality score of $\mathcal{A}$, denoted as $\mathcal{S}(\mathcal{A})$. Now we move on to the next architecture (e.g. in READ, take RNN-dim = 256 and side-net to be GRU) and repeat the above process. After iterating through all architecture candidates, we choose the architecture $\mathcal{A}^*$ that has the best score $\mathcal{S}(\mathcal{A}^*)$, and report the test scores of each best run $\mathcal{R}^*(\mathcal{D})$ of $\mathcal{A}^*$. Therefore, each method in Table 1 adopts the same architectures throughout all datasets. For Full-tuning and BitFit where no architectural hyperparameters are present, we do the learning rate search once to obtain the test scores.

Table 6: Split sizes, training GPU number, and training batch size per GPU node for all GLUE tasks.

|  | CoLA | MNLI | QNLI | MRPC | QQP | SST-2 | STS-B |
|---|---|---|---|---|---|---|---|
| Training Samples (k) | 8.5 | 392.7 | 99.3 | 3.7 | 323.4 | 66.5 | 5.8 |
| Test Samples (k) | 0.52 | 9.8 | 5.4 | 0.2 | 40.4 | 0.9 | 0.8 |
| Validation Samples (k) | 0.52 | 9.8 | 1.0 | 0.2 | 1.0 | 1.0 | 0.8 |
| GPUs | 2 | 8 | 8 | 1 | 8 | 8 | 1 |
| Batch Size per GPU | 48 | 12 | 12 | 96 | 12 | 12 | 96 |

Table 7: Model architectures for four different sized T5 models.

|  | Params (Million) | Encoder Layers | Decoder Layers | Heads | Embedding Dimension | Head Dimension | FFN Dimension |
|---|---|---|---|---|---|---|---|
| $T5_{SMALL}$ | 77 | 6 | 6 | 8 | 512 | 64 | 2048 |
| $T5_{BASE}$ | 248 | 12 | 12 | 12 | 768 | 64 | 3072 |
| $T5_{LARGE}$ | 771 | 24 | 24 | 16 | 1024 | 64 | 4069 |
| $T5_{3B}$ | 2885 | 24 | 24 | 32 | 1024 | 128 | 16384 |

Table 8: Final archtecture choices for all PEFT experiments reported in Section 4.

|  | READ | READ large | Adapter | LoRA | Prompt tuning | LST |
|---|---|---|---|---|---|---|
| Architecture HP Names | RNN-type/ RNN-dim | RNN type/ RNN-dim | Bottleneck size | Rank | Number of prompts | Sidenet-dim |
| Architecture Candidates | {GRU/256, GRU/128, LSTM/128} | {GRU/256, GRU/128, LSTM/128} | {32, 64, 128} | {8, 16, 32} | {10, 20, 30} | {64, 96, 128} |
| Final Choices | GRU/256 | GRU/256 | 64 | 32 | 20 | 96 |

Table 9: Final learning rates for all fine-tuning methods and GLUE datasets

|  | CoLA | MNLI | QNLI | MRPC | QQP | SST-2 | STS-B |
|---|---|---|---|---|---|---|---|
| Full-tuning | $9 \times 10^{-6}$ | $7.16 \times 10^{-5}$ | $3.76 \times 10^{-4}$ | $3.59 \times 10^{-5}$ | $1.75 \times 10^{-4}$ | $4.6 \times 10^{-6}$ | $1.30 \times 10^{-4}$ |
| Adapter | $1.16 \times 10^{-3}$ | $7.47 \times 10^{-4}$ | $4.6 \times 10^{-6}$ | $1.95 \times 10^{-3}$ | $4.6 \times 10^{-6}$ | $1.46 \times 10^{-4}$ | $2.83 \times 10^{-3}$ |
| LoRA | $1.75 \times 10^{-4}$ | $3.05 \times 10^{-5}$ | $9 \times 10^{-6}$ | $1.75 \times 10^{-4}$ | $9 \times 10^{-6}$ | $7.16 \times 10^{-5}$ | $1.15 \times 10^{-4}$ |
| BitFit | $3 \times 10^{-3}$ | $2.83 \times 10^{-3}$ | $2.83 \times 10^{-3}$ | $2.83 \times 10^{-3}$ | $2.83 \times 10^{-3}$ | $3 \times 10^{-3}$ | $2.83 \times 10^{-3}$ |
| Prompt-tuning | $2.83 \times 10^{-3}$ | $1.40 \times 10^{-3}$ | $7.47 \times 10^{-4}$ | $3 \times 10^{-3}$ | $2.83 \times 10^{-3}$ | $3 \times 10^{-3}$ | $2.74 \times 10^{-3}$ |
| LST | $2.51 \times 10^{-4}$ | $1.75 \times 10^{-4}$ | $7.16 \times 10^{-5}$ | $7.47 \times 10^{-4}$ | $3.7 \times 10^{-4}$ | $1.75 \times 10^{-4}$ | $1.45 \times 10^{-3}$ |
| READ | $3.29 \times 10^{-4}$ | $3.67 \times 10^{-4}$ | $1.75 \times 10^{-4}$ | $7.8 \times 10^{-5}$ | $1 \times 10^{-6}$ | $2.5 \times 10^{-6}$ | $4.6 \times 10^{-5}$ |
| READ-large | $8.5 \times 10^{-5}$ | $1.46 \times 10^{-4}$ | $1.75 \times 10^{-4}$ | $1.43 \times 10^{-3}$ | $2.13 \times 10^{-4}$ | $2.04 \times 10^{-4}$ | $7.1 \times 10^{-5}$ |

**Learning rate search** For each learning rate sweep, we do learning rates search in between $[1 \times 10^{-6}, 3 \times 10^{-3}]$ at log-scale for up to 32 rounds, where we employ Bayesian optimization for faster convergence of hyperparameter sweeps at lower computation costs.

**Hyperparameter choices** Table 8 and 9 summarize our final choices of architectural hyperparameters and learning rates.