# OpenReview forum: "READ: Recurrent Adaptation of Large Transformers"
_ICLR.cc/2024/Conference — Submitted to ICLR 2024_

### Official Review · Reviewer_avag · 2023-10-25

**Soundness:** 3 good
**Presentation:** 3 good
**Contribution:** 3 good
**Rating:** 5
**Confidence:** 4

**Summary:**

In this paper, the authors proposed REcurrent ADaption (READ), a lightweight and memory-efficient fine-tuning method for pre-trained foundation models. READ inserts a small RNN network alongside the frozen backbone model, which is trained for downstream tasks. READ can achieve comparable or better accuracy compared to existing parameter-efficient fine-tuning methods on the GLUE benchmark, using less training memory and energy consumption.

**Strengths:**

1. Firstly, the paper is generally well-written and easy to follow.
2. There is no need for an extra step pre-training the side network due to the compact design, making the transfer learning pipeline simple.
3. The proposed method achieves a better accuracy-energy trade-off compared to existing methods on GLUE.

**Weaknesses:**

1. The writing quality should be improved. For example, Appendix?? and Table ??  in Sec 4.
2. The experiments are quite restricted, only using T5 on GLUE benchmarks. It would be better to evaluate more model architectures like GPT-style (if LLaMA is too expensive for the hardware setup, maybe evaluate smaller ones like GPT-2) and more tasks.
3. The latency evaluation in Figure 6 is confusing: why are BitFit and LoRA slower compared to vanilla fine-tuning? Both methods do not introduce extra parameters into the base model (LoRA weights can be fused). It is not reasonable that READ is faster than BitFit/LoRA.
4. What is the non-recurrency setting in Table 2? Why are there more training parameters?
5. The power statistics from NVIDIA's smi are highly unstable. Have you found the calculation to be stable based on a minute-level sampling?
6. One good thing about transformers is the better training parallelism compared to RNN models. Does the design prevent parallel training due to the recursive nature? (I think it can still be parallelized if using the vanilla RNN)

**Questions:**

Please see the questions in the weakness sections. I will wait for the authors' feedback for the final ratings.

---

> ### Author Response · Authors · 2023-11-21
>
> > The writing quality should be improved. For example, Appendix?? and Table ?? in Sec 4.
>
> Thank you for spotting these errors. We will revised the paper to correct these mistakes.
>
> > The experiments are quite restricted, only using T5 on GLUE benchmarks. It would be better to evaluate more model architectures like GPT-style (if LLaMA is too expensive for the hardware setup, maybe evaluate smaller ones like GPT-2) and more tasks.
>
> Thank you for your suggestion, we will run this experiment in the revised version.
>
> > What is the non-recurrency setting in Table 2? Why are there more training parameters?
> The power statistics from NVIDIA's smi are highly unstable. Have you found the calculation to be stable based on a minute-level sampling?
>
> One good thing about transformers is the better training parallelism compared to RNN models. Does the design prevent parallel training due to the recursive nature? (I think it can still be parallelized if using the vanilla RNN)
>
> Thank you for highlighting the significance of a reliable energy consumption comparison. To ensure a fair comparison, we controlled for key factors like training code, optimization algorithm, and hardware across all experiments. Lastly, it's worth noting that the methodology we employ for measuring energy consumption using NVIDIA System Management Interface (i.e. nvidia-smi) is not new, but rather widely recognized and adopted. For example, one can refer to the following papers
>
> [1] Eva García-Martín, Crefeda Faviola Rodrigues, Graham Riley, Håkan Grahn, Estimation of energy consumption in machine learning
>
> [2] Kiran Kasichayanula; Dan Terpstra; Piotr Luszczek; Stan Tomov; Shirley Moore; Gregory D. Peterson, Power Aware Computing on GPUs
>
> [3] Gingfung Yeung, Damian Borowiec, Adrian Friday, Richard Harper, and Peter Garraghan, Towards GPU Utilization Prediction for Cloud Deep Learning
>
> [4] Joseph McDonald, Baolin Li, Nathan Frey, Devesh Tiwari, Vijay Gadepally, Siddharth Samsi, Great Power, Great Responsibility: Recommendations for Reducing Energy for Training Language Models
>
> [5] ROBERT A. BRIDGES, NEENA IMAM, and TIFFANY M. MINTZ, Understanding GPU Power: A Survey of Profiling, Modeling, and Simulation Methods

---

> > ### Comment · Reviewer_avag · 2023-11-22
> >
> > Thanks to the authors for the rebuttal. However, I would like to stick to my previous ratings without further evidence.
> >
> > - Without further evidence, it is hard to demonstrate the generalizability of the proposed method
> > - There lack a response to my question 3, 4, 6

---

### Official Review · Reviewer_UGG1 · 2023-10-31

**Soundness:** 2 fair
**Presentation:** 2 fair
**Contribution:** 2 fair
**Rating:** 5
**Confidence:** 3

**Summary:**

This paper presents READ (Recurrent Adaption), a light weight and memory-efficient finetuning method by inserting a small RNN network to bypass the back propagation to the large backbone network. Results show that READ can achieve great reduction in both memory and energy consumption.

**Strengths:**

(1) Clear description of the proposed method in Section 2.

(2) Good evaluation of the proposed method on T5 model and GLUE benchmark.

**Weaknesses:**

(1) The main result described in abstract/introduction (memory consumption reduced by 56%, and gpu energy reduced by 84%) are compared with full-tuning, not the SOTA memory/energy-efficient finetuning methods. At least we need another data point: a direct comparison to the current best SOTA. From the figure 2, I believe that the comparison with SOTA will lead to much smaller reduction percentages.

(2) A drawback in figure 2 is that it’s hard to justify the advantage of the proposed READ method by just a single data point. Ideally, it’d be helpful to have multiple data points for READ, under different energy/memory consumptions, in order to build a “Pareto curve”. In this way, it’d be much easier to tell whether READ advances the whole Pareto frontier.

(3) In Table 1, it is unfair to only try the proposed method on the larger T5-large. You need to also try other methods + larger T5 in order to prove your points.

(4) As the authors also mentioned in limitation section, it’d be great to add evaluation on GPT-style models and corresponding downstream tasks.

(5) To me, the overall proposed idea make sense but not very exciting. It seems more like an incremental work compared to existing methods (the RNN being the only main novelty).

**Questions:**

See the concerns I listed in weaknesses.

---

> ### Author Response · Authors · 2023-11-21
>
> (1) In Figure 1 and 2, we directly compare READ to other PEFT methods. The reduction percentage is still around 30% compared to LoRA and Adapter in terms of training memory.
>
> (2) Thank you for your suggestion, we will run this experiment in the revised version.
>
> (3) Given the same memory constraint, one can use read to fine-tune a larger model. We show the comparison between the base models and READ on T5 large to show the benefit of READ on the same memory constraint.
>
> (4)  Thank you for your suggestion, we will run this experiment in the revised version

---

### Official Review · Reviewer_BXAc · 2023-10-31

**Soundness:** 3 good
**Presentation:** 1 poor
**Contribution:** 2 fair
**Rating:** 5
**Confidence:** 3

**Summary:**

This paper proposes a efficient adaptation method, recurrent adaption (READ), for Transformers.  The idea is to use recurrent neural networks plus the so-called Joiner networks to learn the correction for the output of the original transformer backbone during fine-tuning.  The experiments verify the effectiveness of the proposed approach.

**Strengths:**

- According to the experimental result presented in this paper, the saving in memory consumption and energy usage are decent.

- The idea of learning _corrections_ for the output of the original Transformer is well-motivated and looks interesting to me. That being said, given my limited familiarity with existing methods in this domain, I'm unable to ascertain the novelty of this idea.

- The authors are upfront about the limitation of the current paper in Sec. 6.

**Weaknesses:**

- I find the notion of recurrence in this paper somewhat tricky. Typically,  Recurrent neural networks process the input sequentially by taking one token as the input at a time and updating the hidden state. However, in this paper, recurrence does not occur in the sequence level. Instead, it occurs at the model layer level, which seems non-standard for me. I highly recommend the authors to clarify this to avoid confusions.

- The scope of the method and experiment seems narrow. It only considers the encoder-decoder Transformers on a single natural language understanding benchmark. I believe the paper will be strengthened significantly by introducing more Transformer settings (e.g., decoder only) and/or more benchmark datasets (e.g., other NLP/vision tasks).

- Training time is not reported in this experiment. I believe it is important to add this Information for a comprehensive comparison among different methods.

- Currently the presentation of the paper is flawed. There are many formatting issues which impedes readability.
  - In the abstract on OpenReview, Latex commands (`\textbf`) are not deleted. $\\%$ is missing in the discussion of the reduction in memory consumption and GPU energy usage.
  - The format of citation is bad in this paper. The authors should have used `\citet` and `\citep` appropriately.
  - In the abstract, the first letter of "Transformer" is capitalized but in the main body of the paper it's not. The authors should make this consistent and I recommend to always capitalize the first letter of "Transformer".
  - Missing references. "Appendix ??" and "Table ??" in Sec. 4.
  - Bad notation. Eq. (4) uses $\bar U$ to denote a cumulative sum. This can be misleading because people are more likely to interpret $\bar U$ as an average.

- Minor comments.
  - Abstract. "empirical evaluation of the GLUE benchmark" $\to$ "empirical evaluation on the GLUE benchmark".
  - Page 4. "the following equation systems gives" $\to$ "the following equation system gives"
  - Page 4. "." is missing in the last sentence of the last paragraph.

**Questions:**

Please refer to the **Weaknesses** part.

One more clarification question:
- What does "normalized" mean in the caption of Fig. 1 mean? What are the original results and how are they normalized?

---

> ### Author Response · Authors · 2023-11-21
>
> ### Notion of Recurrency
> We believe that using the recurency at the model hidden state level is the key behind our method effectiveness. By operating at the model hidden state level, READ does not grow the number of trainable parameters as model sizes increase like other methods, lowering inference cost and fine-tune cost compared to other PEFT methods. The recurrence nature of RNN allows information to flow from one layer to the next, which is why READ outperforms other PETL methods. Last, our
> fine-tuning is transformer-free as only RNN and Feed-Forward Network (FNN) structures are used in READ and require no transformer or attention mechanism.
>
> ### Scope of experiments and method
> We appreciate the reviewer's suggestions on extending READ to more tasks. In response to the reviewer’s valuable insights, we have extended our experimentation to include the MSCOCO dataset using T5-clip model. Notably, our findings remain consistent: READ consistently outperforms most PEFT baselines, while nearly matching the performance of the Adapter, with 68% reduction in training memory consumption. This is highlighted in the provided table below
>
> |               | Full  | Adapter | LoRA  | Prompt | BitFit | READ  |
> |---------------|-------|---------|-------|--------|--------|-------|
> | CIDEr         | 91.9  | 83.4    | 69.5  | 65.2   | 33.6   | 81.7  |
> | Train memory  | 21.20 | 13.38   | 8.16  | 8.39   | 15.77  | 3.88  |
>
>
> ### Training Time
> To provide a comprehensive understanding, we include an analysis below to show the mean and standard deviation for the sums of GPU energy, epochs to convergence, and training time across all GLUE tasks. While this analysis does reveal some variations in energy/time levels, they are not significantly substantial to alter the general trend, as READ continues to stand out as the most energy-efficient approach, with faster convergence than most baselines except for full-tuning. We believe this further supports the merits of our proposed method, and we will include this detailed analysis and clarification in our revised paper.
>
> |              | Full     | Adapter | LoRA   | Prompt | BitFit | LST     | READ  |
> |--------------|----------|---------|--------|--------|--------|---------|-------|
> | Energy (kWh) | 12.52| 6.99 | 10.58| 6.45| 7.68| 10.59 | 2.06|
> | Time (mins)  | 472.74| 485.25| 409.41| 315.35| 291.89| 984.55| 155.11|
>
> ### Normalization in Figure 1
> Normalized here means to the normalized energy consumption relative to full-tuning on GLUE tasks. With full tuning as 1.0 and the PEFT methods as a fraction of full tuning energy consumption.

---

> > ### Comment · Reviewer_BXAc · 2023-11-23
> > **I maintain my original rating**
> >
> > I thank the authors for the response. Some of my concerns remain:
> > - For the notion of recurrency in this paper, I understand the authors' argument, but I think this should be have been made clearer in the paper presentation because this is a non-standard use of RNN. Current writing confuses not only me but also Reviewer avag, and I notice that Reviewer avag feel that the concerns on recurrency are not addressed.
> > - There is no response regarding the presentation issues and typos. Errors are not fixed in the paper.
> >
> > Given these reasons, I maintain my original rating.

---

### Meta-Review · Area_Chair_hBYF · 2023-12-10

**Metareview:**

This paper proposes a parameter-efficient finetuning method for large transformers, which inserts a lightweight RNN network alongside a frozen backbone model to eliminate any needs of back-propagating through it. The proposed method is validated with the T5-large model on GLUE benchmark and is shown to achieve significant reduction in memory and energy compared to the full-finetuning as well as the baseline parameter-efficient transfer learning methods.

All reviewers find the proposed method to be simple yet effective and are impressed by the accuracy-efficiency tradeoffs provided by it. However, all reviewers leaned toward rejection due to multiple reasons:
- Lack of experimental results with models other than T5 on other benchmarks.
- Missing analysis on training time.
- Lack of discussion on the role of recurrence in the proposed method and any limitations it brings.
- Some presentation issues, such as unclear statements and grammatical errors.

The authors provided responses to some of the points raised by the reviewers, but did not faithfully address them all, and the reviewers maintained their ratings afterwards despite the promising aspects of the work. I also agree with the reviewers that there needs to be more experimental validations and in-depth discussions of the methods as well as some efforts in improving the overall presentation, and the paper may benefit from another round of revision.

**Justification For Why Not Higher Score:**

- More experiments should be done with other backbone models (e.g. GPT or LLaMa) to validate the general applicability of the method.
- Needs more in-depth discussions on the role of recurrence and on whether it negatively impacts the parallel computation.
- Presentation should be significantly improved.

Also, the authors did not actively engage in the discussion and did not address all the concerns raised by the reviewers during the rebuttal period.

**Justification For Why Not Lower Score:**

N/A

---

### Decision · Program_Chairs · 2024-01-16

Reject